# A Narrative Review of Non-Pharmacological Strategies for Managing Sarcopenia in Older Adults with Cardiovascular and Metabolic Diseases

**DOI:** 10.3390/biology12070892

**Published:** 2023-06-21

**Authors:** Theocharis Ispoglou, Oliver Wilson, Deaglan McCullough, Luke Aldrich, Panagiotis Ferentinos, Gemma Lyall, Antonios Stavropoulos-Kalinoglou, Lauren Duckworth, Meghan A. Brown, Louise Sutton, Alexandra J. Potts, Victoria Archbold, Jackie Hargreaves, Jim McKenna

**Affiliations:** Carnegie School of Sport, Leeds Beckett University, Leeds LS6 3QQ, UK

**Keywords:** aerobic training, behaviour change, cardiovascular disease, metabolic diseases, nutrition, obesity, resistance training, sarcopenia

## Abstract

**Simple Summary:**

This article explores the relationship between sarcopenia, cardiovascular disease and metabolic diseases. The authors suggest non-drug treatments such as exercise, dietary modifications and behavioural interventions as potential strategies to help older adults manage these conditions. This review highlights the importance of combining strength and aerobic training and adopting comprehensive nutritional strategies. Additionally, the authors propose integrating behavioural science to help people adopt these lifestyle changes. Further research is needed to determine the most effective treatments and ways to make these changes easier for people to adopt. Overall, a comprehensive approach is required to address sarcopenia in older adults with cardiovascular and metabolic diseases.

**Abstract:**

This narrative review examines the mechanisms underlying the development of cardiovascular disease (CVD) and metabolic diseases (MDs), along with their association with sarcopenia. Furthermore, non-pharmacological interventions to address sarcopenia in patients with these conditions are suggested. The significance of combined training in managing metabolic disease and secondary sarcopenia in type II diabetes mellitus is emphasized. Additionally, the potential benefits of resistance and aerobic training are explored. This review emphasises the role of nutrition in addressing sarcopenia in patients with CVD or MDs, focusing on strategies such as optimising protein intake, promoting plant-based protein sources, incorporating antioxidant-rich foods and omega-3 fatty acids and ensuring sufficient vitamin D levels. Moreover, the potential benefits of targeting gut microbiota through probiotics and prebiotic fibres in sarcopenic individuals are considered. Multidisciplinary approaches that integrate behavioural science are explored to enhance the uptake and sustainability of behaviour-based sarcopenia interventions. Future research should prioritise high-quality randomized controlled trials to refine exercise and nutritional interventions and investigate the incorporation of behavioural science into routine practices. Ultimately, a comprehensive and multifaceted approach is essential to improve health outcomes, well-being and quality of life in older adults with sarcopenia and coexisting cardiovascular and metabolic diseases.

## 1. Introduction

Worldwide, the main reason for all-cause mortality is attributed to non-communicable diseases, namely, cardiovascular disease (CVD) and metabolic diseases (MDs) [1]. CVD refers to conditions affecting the heart and blood vessels, such as coronary artery disease, heart failure and hypertension [2]. MDs refer to disorders of processing nutrients and the use of energy. These conditions can include diabetes, obesity and metabolic syndrome, among others [3]. Both CVD and MDs can have serious consequences, including heart attacks, stroke and organ damage [2,3]. CVD remains the leading cause of death worldwide [4]. Moreover, a recent report by the European Society of Cardiology demonstrated that among 56 countries, the prevalence of CVD was accompanied by a substantial economic cost, and the rise in MDs such as obesity and diabetes represented the biggest challenge in further reducing the CVD burden [5]. CVD and MDs are often caused by a combination of genetic, lifestyle and environmental factors [6,7] with older adults being more susceptible to CVD and death [8]. Considering that health behaviours, such as physical inactivity and poor nutrition, are linked to poor outcomes in CVD [7], it is important to manage these conditions through everyday lifestyle changes.

Sarcopenia is another highly prevalent condition in older adults; it is recognised as a muscle disease [9] and is characterised by the loss of muscle mass, strength and function [10]. Although sarcopenia is largely attributed to ageing (i.e., primary sarcopenia), other causes (e.g., disease and physical inactivity) may contribute to sarcopenia (i.e., secondary sarcopenia) [10]. Furthermore, in the revised consensus of the European Working Group on Sarcopenia in Older People 2, acute sarcopenia (likely due to acute illness or injury) and chronic sarcopenia (likely due to long-term conditions such as inflammatory diseases) are identified as new subcategories meriting further attention when designing interventions to better manage muscle-related diseases [10]. Sarcopenia can lead to decreased mobility, increased frailty, increased risk of falls and fractures and other harmful outcomes such as functional decline, increased hospital stay and mortality [11,12]. Furthermore, sarcopenia is associated with poorer mental health outcomes such as depression [13]. Although the pathogenesis of sarcopenia is not fully understood, it is considered a multifactorial process, involving a reduction in hormones that help with muscle mass maintenance, decreased physical activity, anabolic resistance (i.e., changes in the body’s ability to use proteins to build and maintain muscle tissue), nutritional status, inflammation, mitochondrial dysfunction [14,15] and the presence of diseases (e.g., systemic diseases such as diabetes and metabolic syndrome) that can further affect muscle mass and strength [16]. To prevent or manage sarcopenia, it is important to engage in regular physical activity and eat a balanced diet that includes enough protein to support muscle growth and maintenance [17]. Recent evidence has also highlighted a longitudinal association between a higher incidence of CVD with sarcopenia and probable sarcopenia in middle-aged and older adults [18]. Therefore, any efforts to prevent and better manage sarcopenia may be an effective means for reducing the incidence of CVD and improving health and quality of life in older adults with CVD and/or MDs. Collectively, the evidence presented above suggests that sarcopenia and CVD/MDs are associated with ageing, while modifiable risk factors such as a sedentary lifestyle and an unhealthy diet increase the risks associated with these conditions. Treating and managing most cases of sarcopenia—a concern for many clinical and non-clinical groups—may involve lifestyle changes to optimise physical activity levels and nutritional intake, benefitting patients with CVD and MDs by reducing health complications and improving overall health. Risk factors for sarcopenia are depicted in Figure 1.

As sarcopenia is multifaceted, it affects multiple physiological systems, meaning that insights and perspectives can be drawn from different disciplines to provide a comprehensive and holistic approach to treatment. For example, physical activity interventions may be ineffective in older populations simply because older adults might not be able to adhere to these interventions due to a host of factors that may include physical limitations, chronic illnesses, obesity, a lack of social support and low socioeconomic status [19]. Similarly, the effectiveness of nutritional interventions may be negatively influenced by further factors, including underdeveloped professional support, physical restrictions, unhelpful family influences and difficulties in changing eating habits [20]. Therefore, it is important that bespoke interventions, designed to cater to the specific needs of any older population, integrate powerful elements of behavioural modification and implementation aspects to enhance intervention effectiveness. Importantly, every approach may vary depending on the individual’s specific needs and circumstances, as much as it can be influenced by the availability of resources and the preferences of healthcare teams [21,22].

The primary aims of this narrative review are twofold: (a) to delineate the mechanisms that contribute to the pathogenesis of CVD/MDs, with an emphasis on the shared risk factors and reciprocal relationship between sarcopenia and CVD/MDs, and (b) to propose non-pharmacological interventions, such as physical activity, dietary modifications, behavioural interventions and implementation strategies, as potential approaches to mitigate sarcopenia in patients with CVD/MDs.

## 2. Physiological Factors That Underpin Sarcopenia, Cardiovascular and Metabolic Diseases

### 2.1. The Role of Mitochondria in Cellular Processes and Substrate Utilisation and Its Implications for Sarcopenia, Insulin Resistance and Age-Related Muscle Dysfunction

Metabolic diseases are often underpinned by the interrelated pattern of metabolic abnormalities such as atherogenic dyslipidaemia, insulin resistance (hyperglycaemia and hyperinsulinaemia), abdominal obesity and inflammation [23,24]. In this following section, the potential mechanisms that intrinsically link skeletal muscle dysfunction, MDs and ageing are explored.

Mitochondria are crucial for diverse cellular processes, including substrate utilisation, calcium homeostasis, cell proliferation, quality control (e.g., fussion, fission and mitophagy) and apoptosis [25,26]. Mitochondrial dysfunction compromises cell homeostasis and exercise capacity and is a central mediator in the pathogenesis of sarcopenia [25]. However, the molecular basis of the relationship between mitochondrial dysfunction and sarcopenia is multifactorial and poorly understood. Human muscle biopsies have established that the mitochondrial content, measured directly using transmission electron microscopy, is lower in the skeletal muscle of older individuals than in young individuals [27,28], possibly due to differences in the size [29] or number of (subsarcolemmal) mitochondria [28]. As peak oxygen uptake is strongly related to skeletal muscle mitochondrial content in older individuals [30], a reduced content may restrict the performance of daily living tasks. Mitochondrial enzyme activities in β-oxidation (e.g., β-hydroxyacyl-CoA dehydrogenase), tricarboxylic acid (e.g., citrate synthase) and respiratory chain pathways (e.g., succinate dehydrogenase and cytochrome C oxidase) are also lower in the skeletal muscle of older individuals than in young individuals [28,31,32,33]. These findings are accompanied by a lower in vivo oxidative capacity [27,34] and the uncoupling of adenosine triphosphate resynthesis to oxygen consumption [35]. The dual inferiorities of lower mitochondrial content and oxidative capacity are supported by a reduced expression of genes in the tricarboxylic acid and respiratory chain pathways [36,37], mitochondrial ribosomes [36] and in vivo mitochondrial protein synthesis rates with age [38].

Middle-aged patients with obesity and type II diabetes (TIID) also have lower mitochondrial content and size in their skeletal muscle compared to lean age-matched controls, measured using transmission electron microscopy [39,40,41]. This is accompanied by lower in vitro oxidative capacity and respiratory enzyme activity [39,40,41]. Furthermore, in vivo oxidative capacity is impaired in older patients with TIID [42,43], which is linked to lower basal and maximal mitochondrial respiration compared to age-matched controls [43]. Thus, lower mitochondrial function is not only important in the development of sarcopenia but is also suggested to be a contributing factor to the development of insulin resistance [44]. However, some studies that used transmission electron microscopy [45] or citrate synthase activity as a strong biomarker of mitochondrial content [46] reported no difference in the mitochondrial content between middle-aged and older patients with TIID and controls [47,48]. As a result, it is unclear whether a lower mitochondrial content occurs in human skeletal muscle with insulin resistance. Nevertheless, evidence suggests that the skeletal muscle mitochondrial content is correlated with markers of insulin resistance in ageing [49] and TIID [39,40,41], but causation cannot be assumed. Lower mitochondrial respiratory function in the skeletal muscle of patients with TIID compared with controls has also been reported in some, but not all studies, as previously reviewed [50]. The reason for this discrepancy remains unclear, but it may be related to differences in measurement techniques, the normalisation of the respiratory function to the mitochondrial content and/or the physical activity status of the participants [50].

Mitochondria have numerous functions, including the generation of adenosine triphosphate from the oxidation of carbohydrate and fat. Lipid droplets, which store fatty acids such as triacylglycerol, can be in physical contact with the mitochondrial membrane in the subsarcolemmal and intermyofibrillar regions of human skeletal muscle [51]. The attachment of lipid droplets to mitochondria facilitates the channelling of lipid droplet-derived fatty acids to the mitochondria for oxidation based on the energy needs of the cell [52]. Compared to younger individuals, the intramyocellular lipid droplet size and content are elevated in older adults [28] and in individuals with TIID compared to controls [45,51]. Region-specific lipid droplet morphology may be important in ageing and MDs, as the subsarcolemmal lipid droplet size is larger in older individuals [28] and in patients with TIID compared to controls [45,53]. It is worth noting the excessive accumulation of very large subsarcolemmal lipid droplets in muscle fibres with a low subsarcolemmal mitochondrial content in patients with TIID. This may be important, as larger subsarcolemmal lipid droplets have been linked to reduced insulin sensitivity [45,54].

Increased plasma fatty acid uptake and decreased fatty acid oxidation may contribute to the accumulation of intramuscular lipid droplets [50]. A lower mitochondrial content correlates with lower fasting fat oxidation [41], and a negative association between age and fatty acid oxidation in human skeletal muscle primary myotubes has been reported [55]. This finding is in line with lower rates of whole-body fat oxidation rates observed at rest [56,57] and during endurance exercise at the same absolute and relative exercise intensity in older individuals compared to young individuals [58]. Alterations in substrate utilisation with ageing are distinct from those in obesity and TIID. In individuals with obesity, fat oxidation is lower under fasting conditions compared to lean controls [59]. However, the insulin-mediated inhibition of fat oxidation and a consequent switch towards carbohydrate oxidation are impaired in obesity and TIID [59,60]. Individuals with obesity and impaired glucose tolerance exhibit higher rates of fat oxidation and blunted increases in carbohydrate oxidation when transitioning from rest to exercise compared to controls [61,62]. Mitochondria are thought to be functional components in adapting substrate utilisation to substrate availability and energetic demand, termed metabolic flexibility [63]. However, metabolic inflexibility may occur when there is a mismatch between fatty acid supply, uptake and oxidation in skeletal muscle [64,65], leading to the accumulation of fatty acid metabolites that can cause insulin resistance and potentially mitochondrial dysfunction [66].

### 2.2. The Role of Lipid Metabolism and Adipose Tissue Dysfunction in Metabolic Disorders and Cardiovascular Disease

In healthy adipose tissue, excess energy is primarily stored in subcutaneous adipose tissue. However, lipid storage can also occur in other tissues, including visceral adipose tissue visceral adipose tissue, which can lead to inflammation over time. With dysfunctional adipose tissue as observed with MDs and ageing [67], there may be a lipid spillover, which is deposited as visceral adipose tissue and a variety of organs, including muscle tissue [68]. Mendelian randomisation has demonstrated a causal effect of visceral adipose tissue on MDs (hypertension, heart attack/angina, T2D and hyperlipidaemia) by identifying 102 novel loci (i.e., genes) associated with visceral adipose tissue [69]. Visceral adipose tissue has a higher lipolytic rate compared to subcutaneous adipose tissue, which is attributed to the heightened effect of pro-lipolytic catecholamines and reduced effect of anti-lipolytic insulin [68,70]. Consequently, increased lipolysis in visceral adipose tissue leads to a higher flux of free fatty acids to the liver, ultimately resulting in the elevated synthesis of very-low-density lipoprotein and hepatic insulin resistance [70]. Increased levels of very-low-density lipoprotein and chylomicrons create greater competition for lipoprotein lipase, allowing for greater cholesterol ester transfer protein activity with low-density lipoprotein (LDL). Hepatic lipase has a higher affinity for the now triglyceride-enriched LDL, triggering small-dense LDL (sdLDL) production [71]. The prevalence of sdLDL particles underpins atherogenic dyslipidaemia along with high triglyceride levels, and low levels of high-density lipoprotein cholesterol (HDL-C) and are highly associated with increased CVD risk [23,24].

Lipoproteins come in different sizes, densities and lipid content [72]. Two phenotypes, A and B, have been identified to indicate CVD risk [73]; phenotype A has more large and buoyant LDL (lbLDL), while phenotype B has more sdLDL [72,73], which is associated with low HDL-C, high triglycerides levels and metabolic diseases [74,75]. Numerous epidemiological studies have confirmed the reliability of LDL-C as an indicator of CVD risk and that LDL is a causative factor of CVD [76,77,78]. In cases where there is discordance between the two measures, the LDL particle (LDL-P) number may better estimate the CVD risk than LDL-C [79]. When LDL-P is greater than LDL-C, it shows a greater association with CVD events and markers of metabolic health [79,80]. Circulating sdLDL particles have a higher risk of CVD events compared to larger and buoyant lbLDL particles [72,74] because an sdLDL particle has less cholesterol than an lbLDL particle, which leads to a greater number of sdLDL particles when cholesterol is equal [80].

Elevated triglyceride concentrations are associated with at least a 2-fold increase in CVD and mortality risk [81,82]; however, adjusting for HDL-C levels reduces this risk [76]. Although higher HDL-C levels are associated with reduced CVD risk [76,77], therapies that increase HDL-C do not reduce CVD risk [83,84]. HDL, through reverse cholesterol transport, carries cholesterol to the liver for removal and exerts anti-inflammatory effects on peripheral tissues, including arterial lesions [85,86]. Cholesterol efflux from macrophages may offer atheroprotection and is inversely associated with CVD independent of HDL-C concentrations [87,88]. Therefore, the functionality of HDL rather than HDL-C may have a more causal relationship with CVD [89]. Exercise increases HDL-C concentrations in a dose-response manner and has the potential to improve the anti-inflammatory effects and cholesterol efflux capacity of HDL [90]. Triglycerides alone are unlikely to cause atherosclerosis; however, they may act as a marker for triglyceride-rich lipoproteins (TRLs) rich in cholesterol, which are more efficient contributors to atherosclerosis [91,92]. For example, TRLs can enter the arterial intima and undergo direct phagocytosis by macrophages, contributing to foam cell formation, inflammation and atherosclerotic plaques [91,92]. In addition to being involved in lipid storage, mobilisation and lipoprotein metabolism, adipocytes are also endocrine tissues that release cytokines and adipokines [93]. An increase in visceral adipose tissue and atherogenic dyslipidaemia leads to a pro-inflammatory state characterised by elevated C-reactive protein and tumour necrosis factor-α concentrations that are associated with sarcopenia and MDs [94,95]. Given these mechanisms, practitioners should prioritise interventions that can improve HDL functionality, reduce visceral adipose tissue, and improve atherogenic dyslipidaemia, such as exercise and dietary modifications.

### 2.3. Dysfunctional Cellular Metabolism and Insulin Resistance: Implications for Cardiovascular and Metabolic Diseases

This pro-inflammatory state and elevated supply of free fatty acids to the vascular tissues lead to degenerative changes in the extracellular matrix, leading to endothelial insulin resistanceand dysfunction and ultimately hypertension [96]. Insulin resistance of the endothelial cells increases vascular tone due to the disruption of the balance of vasodilators (nitric oxide, prostacyclin) and vasoconstrictors (endothelin, angiotensin), contributing to vascular stiffness [97] and potentially reducing insulin and nutrient delivery to the muscle fibres [98]. Diminished insulin PI3K/Akt signalling in vascular tissue leads to a reduction in nitric oxide production via blunted endothelial nitric oxide synthase action [99]. Consequently, intracellular Ca2+ concentrations remain elevated in vascular smooth muscle cells, allowing vascular contractions to continue, thus limiting the dilation of vascular tissues and increasing vascular tone [99]. Conversely, endothelin-1 production continues via insulin-dependent mitogen-activated protein kinase signalling, leading to vasoconstriction [100]. A reduction in tissue perfusion, such as in skeletal muscle, can lead to a reduction in insulin and nutrient availability and therefore could have consequences for the maintenance of muscle mass and blood glucose clearance with age [98]. Furthermore, along with mitochondrial dysfunction, insulin resistance and prohypertensive factors also stimulate reactive oxygen species, triggering downstream signalling that leads to vascular dysfunction and endothelium permeability, thus contributing to the development of atherosclerosis [101,102].

Dysfunctional cellular metabolism, characterised by lipotoxicity, inflammation, oxidative stress and mitochondrial dysfunction, is implicated in the development of insulin-resistant tissue [103,104], which is a key feature of CVD and MDs [105,106,107]. In healthy insulin-sensitive tissue, insulin binds to the insulin receptor, thus activating the key multifunctional protein kinase Akt, elevating anabolic signalling and reducing catabolic signalling [108]. The activation of Akt initiates downstream signalling, resulting in GLUT4 trafficking and fusion with the cell membrane, allowing cellular glucose uptake [109]. However, within insulin-resistant tissues, this signalling cascade is diminished due to metabolic disturbances primarily upstream of Akt, resulting in reduced glucose uptake [103,104]. Skeletal muscle is the largest tissue for insulin-induced glucose uptake, accounting for ~75% of insulin-stimulated glucose uptake [110] (although adipose tissue and the liver are also involved) [111]. Insulin resistance in adipose tissue triggers an increase in the production of free fatty acids, which can then be deposited in other tissues such as muscle and the liver [112]. Excessive intramyocellular lipid accumulation leads to the build-up of diacylglycerol and ceramide [66], impairing insulin signalling and reducing glucose uptake, glucose oxidation and glycogen synthesis while diminishing its anti-catabolic effects [104]. In summary, excessive fat storage in muscle cells disrupts normal signalling pathways, leading to reduced insulin sensitivity and impaired glucose utilisation. An increase in the supply of free fatty acids can also stimulate liver steatosis, resulting in atherogenic lipoprotein metabolism and diminished insulin stimulation causing hepatic glucose production, further increasing hyperglycaemia [113]. This in turn exacerbates hyperglycaemia, resulting in advanced glycated end products, further stimulating inflammation and the development of MDs [114]. Ultimately, peripheral insulin resistance causes a myriad of metabolic disturbances, triggering a vicious cycle of lipotoxicity, inflammation, oxidative stress and further insulin resistance. These processes are associated with ageing and negatively impact skeletal muscle mass and quality, exacerbating MDs and reducing quality of life.

### 2.4. The Role of Mitochondrial Dysfunction and Reactive Oxygen Species in Obesity-Induced Insulin Resistance and Ageing

Obesity is associated with increased mitochondrial reactive oxygen species (ROS) emission, whereas even a single high-fat meal increases ROS emission in lean healthy controls, suggesting the involvement of an elevated supply of free fatty acids in ROS production [115]. The chronically elevated production of ROS by mitochondria and other sources in skeletal muscle is associated with mitochondrial dysfunction in ageing [116] and obesity-induced insulin resistance [117], while ROS production and oxidative damage remain low in healthy individuals. However, ROS levels can be elevated when substrate delivery to the mitochondria is high, while mitochondrial adenosine triphosphate resynthesis rates are low, such as when a high-fat diet and a physically inactive lifestyle are combined [115]. Elevated ROS levels, beyond the capacity of endogenous antioxidants to scavenge the ROS, cause oxidative damage to mitochondria DNA (mtDNA) and mutations in mtDNA coding regions for respiratory chain proteins, leading to defects in respiratory chain function, the uncoupling of oxidative phosphorylation and further increases in ROS levels in a feedforward mechanism [117,118]. High ROS levels lead to the destruction of mitochondria and, if spread across the mitochondrial reticulum, cell death [119].

### 2.5. The Importance of Physical Activity and Peroxisome Proliferator-Activated Receptor Gamma Coactivator 1α in Maintaining Mitochondrial Quality Control and Skeletal Muscle Health

Mitochondrial quality control in skeletal muscle involves biogenesis, fusion (exchanging components within healthy mitochondria to form an extended mitochondrial network) and fission (removing components), ultimately impacting skeletal muscle mass [120]. Peroxisome proliferator-activated receptor-gamma coactivator 1α (PGC-1α) is a key regulator of mitochondrial biogenesis and function and plays a role in fission/fusion [121]. PGC-1α transcription networks are also positively associated with appendicular lean mass index and hand grip strength [36]. Consequently, a sufficient intramyocellular PGC-1α content is important for skeletal muscle health and protection against sarcopenia [116,121]. In physically active individuals, ageing alone does not decrease the gene expression or protein content of PGC-1α, mitochondrial fission, or fusion proteins [122,123]. Sarcopenia of human skeletal muscle is associated with the downregulation of PGC-1α messenger ribonucleic acid and transcription signatures for PGC-1α networks in fission and fusion compared with controls [36,124]. Similarly, the downregulation of PGC-1α messenger ribonucleic acid [125] and lower fission and fusion protein contents have been observed in the skeletal muscle of patients with TIID compared with controls [126]. These findings highlight the crucial role of PGC-1α in maintaining skeletal muscle health and function and suggest its potential as a therapeutic target for age-related and metabolic disorders. Therefore, a reduction in PGC-1α and an impairment of mitochondrial fission and fusion processes are potential contributors to both sarcopenia and MDs.

A common factor in many studies of ageing, mitochondrial dysfunction and MDs in humans is the participant’s habitual physical activity status; for example, masters athletes have a greater mitochondrial content, oxidative and respiratory enzyme gene expression and activity and in vivo respiratory capacity relative to controls [30,123]. Consequently, master athletes have a well-maintained mitochondrial content and function despite their older age. Moreover, physically inactive elderly individuals retain the ability to adapt to exercise training. It is well known that muscle mass and strength can increase in the elderly following resistance-type exercise training [127]. Mitochondrial content and function also increase in older individuals following endurance-type exercise training [30,128]. This can increase the capacity to oxidise fat during exercise [129], contributing to a reduction in visceral fat mass [130] and chronic low-grade inflammation following exercise training [131]. PGC-1α [30] and mitochondrial fission and fusion proteins are also elevated in masters athletes, suggesting improvements in mitochondrial quality control processes following exercise training [123]. Together, the chronic responses of skeletal muscle to exercise training contribute to improved insulin sensitivity in older individuals [132]. Consequently, physical inactivity, rather than ageing, is a major contributor to the age-related decline in muscle quality [30]. Exercise is strongly indicated to improve skeletal muscle quality in older individuals to protect against the development of mitochondrial dysfunction, sarcopenia, CVD and MDs.

## 3. Exercise and Physical Activity Interventions

Secondary sarcopenia, associated with chronic diseases, requires multiple treatment targets. Physical inactivity, a modifiable risk factor, contributes to both the development and progression of CVD, yet it is often triggered by CVD symptoms such as dyspnoea, angina and claudication upon exertion. Lifestyle modifications, including physical activity, are important for providing non-pharmacological treatments to manage these complex disease states.

Progressive, resistance-based exercise training (RT) has been recommended by the International Clinical Practice Guidelines for Sarcopenia (ICFSR) to treat sarcopenia and improve muscle strength, muscle mass and function [133]. This approach has recently been supported by Hurst et al. [134]. However, the ICFSR review also highlighted the scarcity of high-quality evidence RT interventions in sarcopenic populations [133]. Further research is necessary to understand the impact of RT interventions on chronic CVD and sarcopenia, which often coexist and may share multiple pathogeneses [135,136,137].

Furthermore, the aim of exercise interventions in patients with chronic or congestive heart failure (CHF) is not only to elicit improvements in skeletal muscle parameters but also to improve the chronic disease causing or contributing to the development of sarcopenia while also improving poor overall health and quality of life [138]. High repetition, low-weight RT has been recommended for older adults to improve muscle strength [139]. In older adults, who are inexperienced in RT, significant positive adaptations have been observed in both the skeletal muscle and neuromuscular systems [140]. Additionally, aerobic-based exercise training (AT) has been proposed as a beneficial intervention for sarcopenia due to its capacity to promote mitochondrial biogenesis and insulin sensitivity while reducing oxidative stress [141].

### 3.1. The Potential of Combined Resistance-Based and Aerobic-Based Exercise Training for Improving Muscle and Metabolic Health in Patients with TIID and Sarcopenia

Combining RT and AT could be beneficial for patients with TIID and secondary sarcopenia by targeting both conditions simultaneously. For example, Tan at al. [142] combined RT and AT for a duration of six months to produce significant improvements in muscle strength and function with additional beneficial reductions in fasting insulin, glucose and glycosylated haemoglobin (HbA1c). In another study, interval AT improved lower limb muscle strength and reduced HbA1c levels [143]. More recently, RT interventions using sandbags have provided support for the effectiveness of RT in improving glycaemic control in TIID patients with sarcopenia by reducing HbA1c levels and improving upper-body muscle strength and lean mass [144]. The improved glycaemic control can be attributed to the increase in the glucose transporter 4 (GLUT4) content, which enhances insulin transport and sensitivity [145]. Moreover, RT has also been shown to increase insulin-like growth factor 1, which plays a crucial role in muscle generation, muscle mass and muscle strength [146] and also affects insulin sensitivity. Systematic reviews have also indicated that RT can improve metabolic health in TIID patients without sarcopenia [147]. A recent meta-analysis has demonstrated that a combination of RT and AT resulted in the greatest decrease in HbA1c in TIID patients, followed by either RT or AT independently [148]. Therefore, the combination of RT and AT may be the most suitable intervention for metabolic disease such as TIID. However, it is essential to note that diet-induced weight loss may have the most pronounced and positive effects on metabolic and musculoskeletal health [149]. In addition, nutritional support is necessary to maintain protein intake and prevent the deterioration of musculoskeletal health due to calorie restriction.

### 3.2. The Potential Benefits of Resistance and Aerobic Exercise Training for Cardiovascular Patients with Secondary Sarcopenia: Evidence from Congestive Heart Failure, Coronary Artery Disease, and Peripheral Artery Disease Studies

While AT is often recommended to treat CVD, there is a lack of research on the most suitable exercise interventions when CVD coexists with secondary sarcopenia. CHF is a late-stage condition that affects approximately 2% of the population and is characterised by significant reductions in skeletal muscle leading to sarcopenia; this reduction in muscle mass is often accompanied by reduced cardiorespiratory fitness, increased disease severity and a poor overall prognosis [150,151]. A recent systematic review of exercise-based rehabilitation in CHF patients revealed that RT alone improves lower and upper extremity muscle strength, peak oxygen uptake and 6-min walking distance, without any detrimental effects on the left ventricular parameters, when compared to usual care or an AT intervention [152]. Hence, RT may serve as a feasible strategy for improving physical function and quality of life in CHF patients who might not have the ability or willingness to participate in aerobic activities. It is important to mention that the aforementioned systematic review did not evaluate the occurrence of sarcopenia. Nevertheless, considering the age and multi-morbidities present within such a population, it is plausible to assume the presence of reduced muscle performance, muscle mass and muscle quality. An early study comparing older women with CHF to age-matched controls found that CHF patients had lower muscle strength but similar aerobic capacity [153]. Resistance training improved muscle characteristics and exercise performance in CHF patients, with skeletal muscle adaptations being the main contributor. Supporting the exercise prescription principle of specificity, this supports the idea that muscle function is more specifically affected than aerobic capacity in older CHF patients. In the same study, no improvements were observed in aerobic exercise capacity (peak oxygen uptake and minute ventilation), regardless of significant increases in citrate synthase activity, suggesting that specific AT incorporated as part of a combined exercise intervention may be more suitable for CHF patients. A subsequent study provided evidence that only a combined RT and AT programme led to improvements in both peak oxygen uptake and peripheral muscle strength, which are significant factors in the outcomes and quality of life of patients with CHF [154]. This idea has been reinforced in studies on heart failure models in both humans and rats [154,155,156], which have demonstrated improvements in muscle strength, cardiorespiratory fitness and overall quality of life. Heart failure has also been shown to cause a shift from fast- to slow-twitch muscle fibres, muscle atrophy, oxidative stress, reduced mitochondrial density, increased susceptibility to fatigue and increased biomarkers involved in the ubiquitin-proteasome system, which breaks down muscle proteins [135,157,158]. However, in both rat and human models, AT [157,158] and interval training [156] interventions alleviated the negative physical and molecular changes associated with CHF, thus preserving muscle function. High-intensity interval training (HIIT), in particular, has been linked to an increase in the cross-sectional area of both type I and II muscle fibres, indicating its potential benefits for patients with CHF, as per Bowen et al. [156]. According to Saitoh et al. [159], the mechanism underlying these desirable and protective adaptations involves the increased activation of PGC-1α, which is supported by Lira et al. [160].

Sarcopenia has also been linked to major adverse cardiovascular events after surgery in older patients with coronary artery disease [161,162]. However, RT, AT, or a combination of both have been deemed safe and effective for improving muscle health, cardiorespiratory fitness and quality of life in this population [163,164]. Yamamoto et al. [163] showed that RT interventions improved exercise capacity, muscle strength and general mobility in older adults with coronary artery disease in a meta-analysis of 22 trials. Fan et al. (2021) supported these findings but suggested that combined training methods are more beneficial than either AT or RT alone.

Peripheral artery disease typically affects peripheral system arteries and is frequently accompanied by musculoskeletal abnormalities [165] such as muscle weakness, muscle atrophy and denervation of lower limb muscles [166]. In a small randomised controlled trial, which employed strength testing and histological assessments indicated, RT improved muscle strength and brought about beneficial intramuscular adaptations such as an increase in muscle fibre area and capillary density [167]. Furthermore, a recent meta-analysis by Parmenter et al. [168] showed that RT performed at high intensities (~80% 1RM) was associated with greater walking ability in peripheral artery disease patients, while RT at all intensities improved the distance to onset of claudication.

## 4. Nutrition Interventions

According to He et al. [135], exercise, proper nutrition, hormone therapy and medication are effective in preventing and treating sarcopenia and CVD. Addressing sarcopenia through such modalities may prevent muscle mass, strength and function decline, maintain insulin-stimulated glucose uptake and reduce the risk of TIID and CVD. In the following sections, an overview of nutritional strategies is provided.

### 4.1. Optimising Nutritional Interventions for Sarcopenia and CVD/MDs: Importance of Protein Intake and Quality

To prevent and treat sarcopenia in older adults and to address CVD and MDs, nutritional interventions emphasise appropriate energy intake and dietary protein including aspects such as quantity, quality and timing of consumption [17,169]. In older adults, the impaired muscle protein synthesis response to food and physical activity contributes to muscle mass loss and a decline in metabolic health [170]. Adequate energy and dietary protein intake, including amino acids, are necessary to support muscle growth, maintenance and reduce health risks [17,169]. Dietary proteins provide amino acids for muscle protein synthesis and glucose control, resulting in reduced triglyceride levels and potentially improve insulin response to feeding [137]. In individuals with a high body mass index, inadequate protein intake increases the risk of ‘sarcopenic obesity’ [171], which can increase the risk of cardiovascular-related illness, frailty and falls, and it can reduce muscle functionality. When assessing individuals with excess adiposity and other complex metabolic and lifestyle issues, the key focus of exercise and dietary strategies is often overall weight reduction [172]. However, such interventions often compromise the ability to preserve muscle mass and function due to the emphasis on achieving energy deficits. This can create a vicious cycle where older adults become weaker and less able to engage in physical activity, which is crucial for maintaining energy balance and combating obesity.

To optimise skeletal muscle health and reduce the risk of CVD/MDs, it is recommended to exceed the recommended dietary allowance (RDA) of 0.8 g/kg body mass per day for protein intake, as supported by long-term studies [169,173,174,175]. According to the recommendations by the European Society for Clinical Nutrition and Metabolism (ESPEN), individuals suffering from an acute of chronic disease should increase their protein intake by 50–90% (~1.2–1.5 g/kg/bm per day) above the RDA and maintain an energy intake of 25–30 kcal/kg/day [169]. For older adults, a phased protein intake of 25–30 g of protein three times a day is recommended to optimise the rates of muscle protein synthesis (MPS) [17,176,177,178]. This extra intake is needed because older individuals tend to be less sensitive to anabolic stimuli such as protein and prone to ‘anabolic resistance’ [178,179]. Studies show that the absolute protein requirement for the stimulation of muscle protein synthesis within a single meal is greater in older adults (~35 g) than in younger adults (~20 g) [180,181]. However, increasing protein intake may cause unwanted appetite loss and reduce overall nutritional intakes [182]. This has been termed as the ‘protein paradox’ [17], where increasing protein intake may unintentionally cause reductions in energy and protein intakes through partial energy redistribution. Interestingly, older UK adults had low protein intake, with 85% failing to meet the minimum RDA of 1.2 g/kg/day, as per ESPEN, and only 24.6% achieved protein intakes of 25 g per meal according to the Protein 4 Life collaboration [176,183]. Therefore, it is necessary to implement population-wide strategies to boost protein intake for optimal MPS stimulation while considering cardiovascular health, with recent research focusing on protein quality and supplementation.

Branched-chain amino acids such as leucine, isoleucine and valine are effective at regulating hormones involved in food intake and glucose regulation [184,185]. Leucine, in particular, has been studied for its ability to preserve muscle mass without compromising reductions in appetite and energy intake when consumed alongside other essential amino acids (EAAs) [182,186,187,188]. It was demonstrated that in muscle-depleted patients with CHF, EAA supplements combined with adequate energy–protein intake improved the nutritional and metabolic status [189]. However, not all outcomes improved, possibly because the EAAs were given separately from meals, leading to increased oxidation rates, as seen with fast-acting protein sources [190]. Palatability issues when diluting EAAs in water may have also affected the compliance data. Importantly, when considering protein intake in the context of CVD, it is crucial to be mindful of the potential of high-protein diets to include red meat, processed meat and other foods high in saturated fats, which can increase serum lipid levels such as triglycerides and heighten the risk of heart disease [191]. Recent studies show that plant protein intake is associated with reduced all-cause and cardiovascular mortality, whereas animal protein intake may increase the risk of CVD mortality, as seen in previous cohort studies [192,193]. This suggests that protein sources are important for long-term health. Recent studies have also linked higher intakes of total meat and unprocessed red meat to a higher incidence of atherosclerotic cardiovascular disease, possibly due to the plasma levels of metabolites produced by gut microbiota [194]. Therefore, for managing sarcopenia in older adults with CVD/MDs, an effective approach for increasing protein intake may involve substituting animal-based protein with plant-based alternatives.

### 4.2. Incorporating Antioxidant and Anti-Inflammatory Foods for Sarcopenia, Cardiovascular Disease and Metabolic Disease Management and Prevention

Sarcopenia, CVD and MDs share common characteristics of oxidative stress and inflammation [195,196], with ageing increasing oxidative damage and inflammation [197,198]. In contrast, antioxidant and anti-inflammatory compounds in food and drinks can increase the antioxidant capacity and reduce inflammation. Studies [199,200] show that consuming antioxidant-rich foods such as fruits, vegetables, nuts, cacao, oils, micronutrients (e.g., selenium, magnesium, iron and vitamins A, C and E), plant-derived polyphenol-rich foods (e.g., berries and cherries) and long-chain fatty acids (e.g., omega-3 fatty acids) can help manage, treat and prevent sarcopenia in older adults aged >55 years [201]; the most promising interventions were increased fruit and vegetable consumption, tea catechins, magnesium and combined vitamin E, D and protein supplementation. Other studies support these findings, suggesting that minerals may aid sarcopenia prevention and treatment in healthy and frail older adults (>65 years) [202], whereas a higher dietary inflammatory index is associated with worse sarcopenic symptoms in older adults (>65 years) [203]. Although most evidence excludes sarcopenic individuals with specific health conditions (e.g., sarcopenic obesity), a recent study associated diets with a higher total antioxidant capacity with reduce abdominal obesity, fasting glycaemia and sarcopenia in TIID patients [199]. Among 158 healthy older adults, a potential association between increased inflammation (based on C-reactive protein levels) and slightly reduced appetite was identified [204]. It is crucial to note that while reducing oxidation and inflammation can be beneficial, these processes play an important role in physiological adaptations to exercise [205]. Given the importance of physical activity (see Section 3) in managing sarcopenia, CVD and metabolic diseases, interventions with antioxidant or anti-inflammatory foods or drinks should be carefully considered, as excessive levels may hinder exercise benefits [206]. However, some studies show that antioxidant supplementation can improve various outcomes when combined with exercise in certain populations [207,208]. Emerging evidence suggests that omega-3 polyunsaturated fatty acids may benefit sarcopenic older adults due to their anti-inflammatory properties and anabolic effect on muscle by reducing insulin resistance and activating mTOR signalling [209]. Interestingly, older TIID patients with sarcopenia have lower omega-3 fatty acid intake than those without sarcopenia [210]. Although few randomized controlled trials have investigated antioxidant and/or anti-inflammatory intake in sarcopenic individuals with CVD or MDs, existing evidence is promising [211,212]. Korean Red Ginseng at 3 g/day for 24 weeks significantly improved the biomarkers of sarcopenia in older adults with TIID, especially postmenopausal women [213]. Another randomized controlled trial demonstrated that tea fortified with catechins alongside EAA supplementation and exercise for three months improved walking speed, stride length, vitamin D and total body fat mass compared to a control group in women with sarcopenic obesity [214].

### 4.3. The Importance of Vitamin D in Preventing and Managing Sarcopenia, Cardiovascular Disease and Metabolic Diseases in Older Adults

Vitamin D is essential for bone and muscle health; however, deficiency (often considered as serum 25-hydroxyvitamin D <50 nmol/L) affects people of all ages worldwide [215]. Older individuals are particularly susceptible due to factors such as reduced dietary intake, sun exposure, and reduced ability to synthesise vitamin D [216]. Insufficient vitamin D levels predict outcomes associated with reduced quality of life, including the loss of muscle mass, increased fragility, risk of falls and hospitalisation, as well as a loss of independence [217]. It is not surprising that low vitamin D levels are linked with sarcopenia risk [218] and various metabolic diseases [219]. However, evidence is equivocal on the effectiveness of vitamin D supplementation for preventing or treating sarcopenia, partly due to study designs and intervention differences [220]. Interestingly, combining vitamin D with protein (especially sources rich in leucine) appears to confer more consistent positive outcomes in increasing muscle mass, strength and performance in sarcopenic older adults compared to placebo or standard practice [221]. A recent review supported the role of vitamin D deficiency in osteosarcopenic obesity development; however, no trial has assessed the efficacy of supplementation in this population [222]. Vitamin D supplementation has also been suggested for managing sarcopenia in heart failure patients by modulating inflammation, blood pressure and endothelial function [223], but more research is needed. It is important to note that interactions between certain drugs and vitamin D absorption or metabolism have been proposed, including treatments for obesity (e.g., lipase inhibitors) and CVD prevention (e.g., statins) [217]. Further investigation into these interactions is warranted to corroborate their potential impact on sarcopenic symptoms in older adults with CVD and MDs. Nevertheless, given the numerous essential roles of vitamin D, supplementation is recommended, particularly for older adults at risk of insufficient levels. To achieve a serum 25-hydroxyvitamin D level of approximately 75 nmol/L, the International Osteoporosis Foundation recommends a daily intake of 20–25 µg/day or 800–1000 IU/day for older adults [224].

### 4.4. The Gut Microbiota–Skeletal Muscle Axis: Implications for Sarcopenia, CVD and Obesity

Gut microbiota play a crucial role in maintaining health, influencing various physiological and metabolic processes including nutrient digestion and absorption, the production and regulation of short-chain fatty acids, inflammation and immune function and appetite and food intake [225,226]. Gut health has been associated with many clinical conditions and diseases including sarcopenia, obesity and CVD [225,227]. The gut–muscle axis, a relatively new concept, suggests that microbiota can protect against age-related muscle loss and dysfunction [228,229,230]. Differences in microbiota composition are observed between frail, sarcopenic patients and non-sarcopenic controls [231,232]. Gut microbiota dysbiosis, with reduced microbial diversity and beneficial bacteria, is more common in advanced age, similar to sarcopenia and CVD [233]. The health and composition of gut microbiota, which can impact age-related conditions, are influenced by various factors including genetics, biological sex, ethnicity, environment, antibiotic use, physical activity, and diet [225]. To target the gut microbiota–skeletal muscle axis and improve these conditions, strategies such as increasing the intake of probiotics and prebiotic fibre have demonstrated positive effects on the gut microbiome in older adults [234]. Promising links have been observed between increased dietary intakes of probiotics and prebiotic dietary fibre intake and sarcopenia symptoms [226,228,235]. Most evidence in this area relies on rodent/animal models, with human studies mainly focusing on associations between gut health and muscle markers. To our knowledge, no randomized controlled trials have targeted the human microbiome of sarcopenic individuals, and more research is warranted.

## 5. Multidisciplinary Approaches

Interventions designed to manage sarcopenia, CVD and MDs are multi-faceted. It is well documented that physical activity (PA) interventions, particularly progressive, resistance training prescriptions, can have a positive impact on preventing the deterioration of both muscle mass and function in adults aged ≥60 years [236] and amongst people living with CVD [237]. The 2020 World Health Organization [238] and the United Kingdom’s Chief Medical Officers’ PA guidelines [239] have recently updated their recommendations to emphasise new understanding of the importance of muscle strengthening activities. Despite the robustness of the evidence, population adherence rates remain low [240], possibly due to lack of awareness of strength-based guidance for both clinical and non-clinical populations [241] as well as cognitive preferences and capabilities that suggest ‘reluctance’. To address muscle loss, as highlighted in the previous section, nutrition interventions focused on addressing protein deficiencies have been shown to be beneficial [242]; however, other nutrition strategies focus on the consumption of vitamin D or the removal of alcohol and fatty acids from the diet [243]. The next subsection delves into some of the challenges older adults may encounter when trying to engage in both physical activity and nutrition interventions.

### 5.1. Challenges of Engaging in Interventions

Older populations attempting to engage in both PA and nutrition interventions for sarcopenia, CVD and MDs face a myriad of challenges. These challenges may include pain, fatigue, fear of movement and physical harm, lack of social support and lack of motivation among others [244,245]. Additionally, older adults often accept and believe that muscle degeneration is an inevitable part of the ageing process [246] and believe they are too old to learn and benefit from exercise [247]. However, challenging this belief through education on how muscle can be developed at any age through diet and exercise could enhance engagement in PA and protein intake. Another challenge comes from professional attitudes; in the phenomenon of ‘compassionate ageism’, programme staff view older adults as requiring protection from the high level of effort that can be associated with engaging in strength training [240]. Moreover, the desire to avoid vigorous exercise may also stem from a historical perspective of engaging in physical labour, leading to the notion of “going lighter as I get older”. Interestingly, this misconception about avoiding vigorous intensity applies to both aerobic and strength training participation and is highlighted in national PA recommendations and guidelines for older adults [240].

### 5.2. What Are the Possible Ways Address These Challenges?

For interventions to be effective, it is pertinent that practitioners understand and work to overcome known barriers. One way this could be achieved is by modifying behaviour, a topic that has generated increasing attention in recent literature, particularly in relation to individuals with sarcopenia and/or CVD/MDs. Evidence of behavioural change has emerged from recent randomised controlled trials (e.g., [248,249]), systematic reviews (e.g., [250,251]), meta-analyses (e.g., [248,249,252,253]) and studies established to inform programme design and improvement (e.g., [254,255,256]). It is imperative that practitioners understand the underlying principles and mechanisms underpinning intervention design for this population to bring about meaningful changes. When designing interventions, it is important to consider the challenges affecting daily behaviour; limited uptake and engagement may be more to do with the system(s) that support them as with the individual. A multifactor approach is needed. For example, this could involve supplementing individualised processes (e.g., accentuating motivation and capability), with social processes (e.g., involving copying others, conformity and cooperation) and structural factors (e.g., legislation and incentives) [257]. As such, approaches are needed that support both participants and those delivering the interventions. Ting-ru et al. [258] provided insight into the facilitators of PA for those living with sarcopenia and diabetes. The study revealed the motivational power of the perceived benefits of increased vitality, strength and appetite. Furthermore, this study highlighted the significance of recognising the limitations imposed by life circumstances when designing exercise routines. For instance, this could involve modifying the range of activities or adapting them to be done at home, using assistive devices (e.g., an umbrella for walking), to be active with friends, or using a wheelchair to get to a park and to plan routes where walking areas allow easy access and passage despite pre-existing and/or disease-specific limitations.

Van den Helder et al. [259] presented a recent case of an intervention that implemented client-centred behaviour modification to enhance physical performance and counteract the decline in physical function and sarcopenia. This intervention included exercise and protein counselling and resulted in improvements in muscle mass, strength, protein intake, gait strength and physical activity levels. Behaviour modification was based on the principles highlighted by the Medical Council Framework for behaviour change [260] and was personalised by using a mobile application [261]. This incorporated effective behaviour change techniques of goal setting, action planning and self-monitoring, as identified by the CALO-RE taxonomy [262]. Behaviour modification interventions are often based on recent theories and models such as the COM-B and dual-mode approaches to behaviour change [263]. These theories emphasise the importance of both self-regulation (e.g., goal setting) and automatic affective associations (e.g., promoting feelings of pleasure, well-being, connection and control) [264]. It is suggested that interventions underpinned by theory have better outcomes and higher adherence rates [265]. Therefore, when designing interventions, practitioners should strive to incorporate a theoretical foundation to support their approach, emphasising interventions that over-ride the barriers clients identify as being most influential.

Policy makers are increasingly turning to behavioural science for insights into how to improve both decisions and outcomes [266]. For example, a recent “mega” study found that offering small financial incentives, also known as micro incentives, could generate larger treatment and adherence effects and still attain the desired outcome [267]. Equally, experiencing the intrinsic reward of immediate joy and fun alongside evidence of personal progress provides a powerful mix of incentives for prolonging persistence [268] and one that could be considered for this area.

To achieve long-term adherence, incorporating technology is increasingly recommended; in mHealth and eHealth, for example, personalised apps for tablets have proven effective in promoting adherence to PA and diet interventions in older adults [259,269]. Furthermore, an mHealth randomised controlled trial used FitBit trackers to predict 12-month PA engagement and found that the best predictor of engagement was adherence in week one, which was achieved by 66.5% of the participants [270]. Notwithstanding a sample dominated by programme-ready, overweight–obese females (80%), replicating these findings would support reframing the powerful message of ‘every contact counts’ into ‘first week contacts count most’. For this reason, ensuring that individuals feel welcomed, included and valued [271] is critical in creating positive initial experiences within any programme. Moreover, older adults living independently with frailty are more likely to experience heightened social and or physical isolation [272]. Therefore, investing in interventions that can be remotely designed and delivered through technology could be advantageous. This approach supports those living independently, with frailty, who struggle independently outside their own home, or have heightened anxiety of contact with people outside their home [273].

## 6. Conclusions and Future Directions

In conclusion, breaking the vicious cycle of sarcopenia in older adults with cardiovascular and metabolic diseases necessitates a multifaceted approach that encompasses exercise, nutrition and multidisciplinary interventions. The importance of incorporating combined training, comprehensive nutrition strategies and behavioural science integration to address sarcopenia in older adults with CVD and MDs has been highlighted in this review. These interventions are crucial in the prevention and management of sarcopenia and related conditions. Relevant stakeholders are strongly encouraged to implement these evidence-based strategies in their practice to improve patient outcomes (Figure 2). While our review provides a comprehensive summary of the available literature on this topic, incorporating perspectives from various disciplines, it is important to note that due to the scope of our literature search, some relevant studies may have been overlooked. This could potentially impact the comprehensiveness of our review and limit the generalizability of our findings. Future research should focus on high-quality randomized controlled trials to improve exercise and nutritional interventions while also examining the best ways to incorporate behavioural science into everyday practices for better initial adoption and for longer-term success of sarcopenia interventions. Further research is needed to investigate the unique challenges and barriers associated with different disease types, frailty, supervision and resources to facilitate systemic change and individual empowerment. Ultimately, a coordinated effort in these areas can improve health outcomes, well-being and quality of life in older adults with sarcopenia and coexisting cardiovascular and metabolic diseases.

## Figures and Tables

**Figure 1 biology-12-00892-f001:**
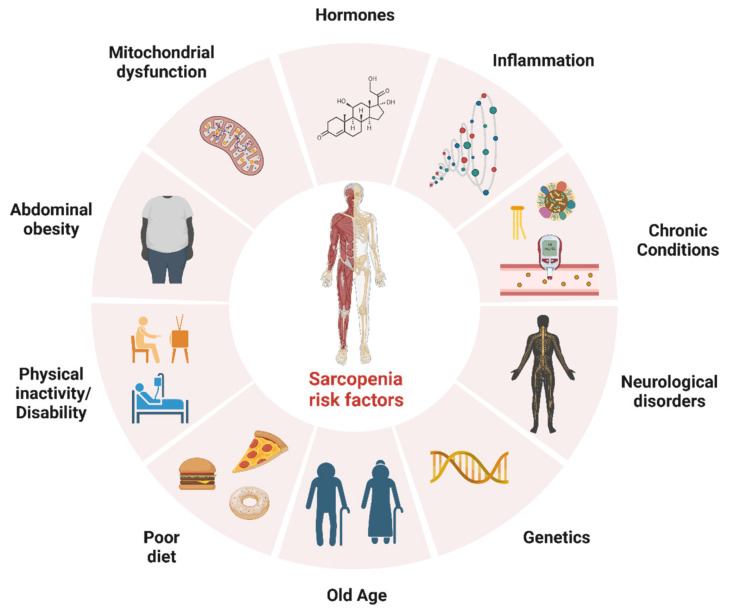
Risk factors for sarcopenia. The figure was created with BioRender.com (Accessed on 1 February 2023).

**Figure 2 biology-12-00892-f002:**
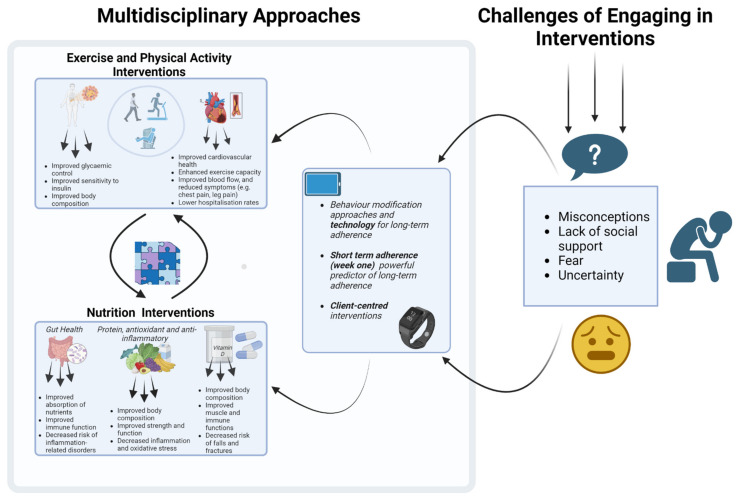
Multidisciplinary approaches to address sarcopenia in older adults with cardiovascular and metabolic diseases. The figure was created with BioRender.com (accessed on 15 March 2023).

## Data Availability

No new data were created or analyzed in this study. Data sharing is not applicable to this article.

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
