# Peer review of "A Narrative Review of Non-Pharmacological Strategies for Managing Sarcopenia in Older Adults with Cardiovascular and Metabolic Diseases"

_biology, 2023, doi:10.3390/biology12070892_

Round 1

Reviewer 1 Report

My main concern is why authors conducted a narrative review rather than a systematic one.  The paper is weak in the current form, here are some indications to improve:

·      The title is unclear, need to be shorten, and the rewording should include “narrative review”

·      In the abstract a background statement should be added before the aim. 

·      The keywords should be ordered in alphabetical order. 

·      The Introduction section is broad and speculative should be drastically shortened and focused not to exceed 1000 words.

·      A short method section should be added as well as a checklist, for example: https://www.elsevier.com/__data/promis_misc/ANDJ%20Narrative%20Review%20Checklist.pdf

·      A result section should be added.

·      A discussion section should be added organised as follows: (i) Main findings; (ii) Strengths and limitations; (iii) Clinical implications; (iv) Future direction for future research.

·      The conclusion section should be very short and concise

·      A table that include the most influent studies and their finding should be added. 

·      The number of figures should be reduced.   

Author Response

Response to Reviewer 1 Comments

Point 1: My main concern is why authors conducted a narrative review rather than a systematic one.  The paper is weak in the current form, here are some indications to improve:

Response 1: Our perspective differs from yours, but we respect your opinion. Many editorial board members would argue that a narrative or scoping review can be just as beneficial as a systematic review, especially when the main emphasis of a paper is not on methods. Excluding articles that do not address the research question is standard for systematic reviews, but concerns emerge when the exclusion criteria are not based on the quality of scientific presentation and interpretation. Factors such as a small sample size or unbalanced design may not warrant a study’s exclusion from the review, and thus, should not justify dismissing the study’s message. In these situations, a systematic review might not be the optimal approach, as it could potentially miss valuable information from relevant studies. In contrast, a narrative review can offer more than just a description of the findings from a systematic approach, by thoroughly exploring various interrelated aspects of a topic area. A notable drawback of the systematic approach is its exclusion of studies based on predetermined criteria. This limitation leads to the question of how to incorporate crucial information from studies omitted by the systematic approach. The primary shortcoming of the systematic approach is its exclusive and selective nature, which can result in losing valuable perspectives from relevant studies. Specifically, we believe our review provides a broad and comprehensive overview of the topic under investigation, which will enable readers to gain a thorough understanding of the subject matter. In addition, incorporating a wide range of studies and perspectives fosters a more multifaceted and inclusive understanding of the topic.

Point 2: The title is unclear, need to be shorten, and the rewording should include “narrative review”

Response 2: We have removed “Breaking the vicious cycle”, and we have incorporated the term “narrative review” in the title, as per your request, to provide clarity.

Point 3: In the abstract a background statement should be added before the aim. 

Response 3: Including a single background statement is not suitable in this case, as the review is not a systematic one.

Point 4: The keywords should be ordered in alphabetical order. 

Response 4: We apologise for the oversight-this has now been rectified.

Point 5: The Introduction section is broad and speculative should be drastically shortened and focused not to exceed 1000 words.

Response 5: The existing introduction is approximately 800 words, which is significantly below your requested length of 1000 words. We recognise your concerns about the introduction’s broad perspectives. However, it was essential to effectively integrate various viewpoints, considering the multidisciplinary nature of this review. We feel that this approach leads to a clearer and more efficient presentation of the subject matter.

Point 6: A short method section should be added as well as a checklist, for example: https://www.elsevier.com/__data/promis_misc/ANDJ%20Narrative%20Review%20Checklist.pdf

Response 6: This approach is not appropriate given that the current review is a narrative one, which brings together perspectives from various disciplines. It would be unrealistic and imprudent to include different search strategies for each topic area explored

Point 7: A result section should be added.

Response 7: For the same reasons mentioned in number 6 above, this approach is not suitable.

Point 8: A discussion section should be added organised as follows: (i) Main findings; (ii) Strengths and limitations; (iii) Clinical implications; (iv) Future direction for future research.

Response 8: For the same reasons mentioned in number 6 above, this approach is not suitable.

Point 9: A table that include the most influent studies and their finding should be added. 

Response 9: For the same reasons mentioned in number 6 above, this approach is not suitable.

Point 10:     The number of figures should be reduced.   

Response 10: This has been addressed by the removal of “Figure 2”

Reviewer 2 Report

This manuscript entitled “Breaking the vicious cycle: non-pharmacological strategies for managing sarcopenia in older adults with cardiovascular and metabolic diseases” was primarily aimed to review the relationship between sarcopenia, cardiovascular disease, and metabolic diseases and suggest the non-drug treatments. The authors bring an interesting study, but there are still some problems that cannot up this article to a publishing level. Suggestions are listed in the specific comments below. In general, the article should be written in the past tense. Besides, there are too many abbreviations which do hinder readability.

Specific comments:

1.     In the Introduction part, line 44, “Metabolic diseases (MDs) refer to disorders of…” Please provide the abbreviation (MDs) in line 42 “ and metabolic diseases such as diabetes”

2.     In 2.2 part, Line 222, “In healthy adipose tissue, excess energy is primarily stored in subcutaneous adipose tissue (SAT).” There are too much abbreviations in this study which influence reading, please consider to delete some unnecessary abbreviations. For example, SAT only appeared twice in this manuscript.

3.     In 2.2 part, Line 222, “high TG and metabolic diseases” What does the abbreviation “TG” mean? Such problem also existed in lint 254, “FFA”.

4.     In 2.5 part, line 334-335, “A factor common in many studies of ageing, mitochondrial dysfunction, and MD in humans is the participant’s habitual physical activity status.” Please cite relevant papers here.

5.     In 5.2 part, line 752, “To achieve long-term adherence, incorporating technology is increasingly recommended.” More references are needed here.

6.     What are the limitations of this study? Please provide relevant description in the discussion part.

7.     In the Conclusion and future directions part, can you briefly conclude exercise, nutrition, and multidisciplinary interventions you recommend in this manuscript?

Extensive editing of English language required

Author Response

Response to Reviewer 2 Comments

Point 1: This manuscript entitled “Breaking the vicious cycle: non-pharmacological strategies for managing sarcopenia in older adults with cardiovascular and metabolic diseases” was primarily aimed to review the relationship between sarcopenia, cardiovascular disease, and metabolic diseases and suggest the non-drug treatments. The authors bring an interesting study, but there are still some problems that cannot up this article to a publishing level. Suggestions are listed in the specific comments below. In general, the article should be written in the past tense. Besides, there are too many abbreviations which do hinder readability.

Response 1: Thank you for your comments. We appreciate your feedback and acknowledge that there are some issues to resolve before the article is ready for publication. In response to your suggestions, we have made an effort to minimise the number of abbreviations to improve readability and have made sure to use the appropriate tense throughout the text when necessary. We have refrained from using future tense, and some of the abbreviations, such as IR, SAT, VAT, VLDL, CRP, FFA, PAD, and CAD, have been removed to enhance readability. It is clear how this approach improves readability so thanks for bringing this to our attention.

Point 2: In the Introduction part, line 44, “Metabolic diseases (MDs) refer to disorders of…” Please provide the abbreviation (MDs) in line 42 “ and metabolic diseases such as diabetes”

Response 2: Thank you- we have made the appopriate adjustments.

Point 3: In 2.2 part, Line 222, “In healthy adipose tissue, excess energy is primarily stored in subcutaneous adipose tissue (SAT).” There are too much abbreviations in this study which influence reading, please consider to delete some unnecessary abbreviations. For example, SAT only appeared twice in this manuscript.

Response 3: Thank you-we agree with you and have made the appopriate adjustments.

Point 4: In 2.2 part, Line 222, “high TG and metabolic diseases” What does the abbreviation “TG” mean? Such problem also existed in lint 254, “FFA”.

Response 4: Thank you- we have replaced TG with full word (triglycerides). Please also refer to our response to your first point above regarding abbreviations.

Point 5: In 2.5 part, line 334-335, “A factor common in many studies of ageing, mitochondrial dysfunction, and MD in humans is the participant’s habitual physical activity status.” Please cite relevant papers here.

Response 5: Thank you for bringing this to our attention. The references were added to the rest of the paragraph. The statement in question was a thematic statement for the paragraph, but we have revised it to avoid any confusion. Please refer to the revised text below. “ A factor common in many studies of ageing, mitochondrial dysfunction, and MD in humans is the participant’s habitual physical activity status; for example, masters athletes have a greater mitochondrial content, oxidative and respiratory enzyme gene expression and activity, and in vivo respiratory capacity relative to controls [34,141,145]”

Point 6: In 5.2 part, line 752, “To achieve long-term adherence, incorporating technology is increasingly recommended.” More references are needed here.

Response 6: Similar to our response to your previous point, this was a thematic statement. We have added a semicolon to address your concern.

Point 7:  What are the limitations of this study? Please provide relevant description in the discussion part.

Response 7: Thank you for highlighting the importance of outlining the limitations of our review. We have included the following text in the “Conclusion and future directions” subsection: “While our review provides a comprehensive summary of the available literature on this topic, incorporating perspectives from various disciplines, it is important to note that due to the scope of our literature search, some relevant studies may have been overlooked. This could potentially impact the comprehensiveness of our review and limit the generalizability of our findings.”   

Point 8: In the Conclusion and future directions part, can you briefly conclude exercise, nutrition, and multidisciplinary interventions you recommend in this manuscript?

Response 8:  Thank you for requesting that we reiterate this important message: The relevant passage in the “Conclusion and future directions” subsection has been amended accordingly: “The importance of incorporating combined training, comprehensive nutrition strategies, and behavioural science integration to address sarcopenia in older adults with CVD and MDs has been highlighted in this review. These interventions are crucial in the prevention and management of sarcopenia and related conditions. Relevant stakeholders are strongly encouraged to implement these evidence-based strategies in their practice to improve patient outcomes.”

Point 9: Comments on the Quality of English Language: Extensive editing of English language required.

Response 9: We have proofread the entire document and made efforts to improve the syntax and grammar where appropriate. We also made sure that correct punctuation was used consistently throughout the text to effectively convey the key messages. We have also refrained from using future tense and first person (i.e., “we”). We have made every effort to present our research in the clearest and most professional manner possible, and we hope that these revisions have improved the quality of our manuscript.

Reviewer 3 Report

The authors present a narrative review entitled: "Breaking the vicious cycle: non-pharmacological strategies for managing sarcopenia in older adults with cardiovascular and metabolic diseases" that explores the relationship between sarcopenia, cardiovascular disease and metabolic diseases, a topic with significant clinical relevance. that must be studied and updated.

In the title of the manuscript, please remove "title:".

The authors reflect figure 1 on the risk factors for sarcopenia, but has this figure been made by the authors? Otherwise, the corresponding bibliographic citation should appear. The same happens with figures 2 and 3.

I do not see the summary subsections that appear in each of the sections as appropriate, please delete.

Otherwise, this manuscript complies with journal guidelines and is suitable for publication after the above changes.

Author Response

Response to Reviewer 3 Comments

Point 1: The authors present a narrative review entitled: "Breaking the vicious cycle: non-pharmacological strategies for managing sarcopenia in older adults with cardiovascular and metabolic diseases" that explores the relationship between sarcopenia, cardiovascular disease and metabolic diseases, a topic with significant clinical relevance. that must be studied and updated.

Response 1: Thank you for your kind comments. We appreciate your recognition of our narrative review, and we share your belief that this topic is of significant clinical importance and requires continual study and updates.

Point 2: In the title of the manuscript, please remove "title:".

Response 2: Thank you for bringing this to our attention. We have addressed your point.

Point 3:  The authors reflect figure 1 on the risk factors for sarcopenia, but has this figure been made by the authors? Otherwise, the corresponding bibliographic citation should appear. The same happens with figures 2 and 3.

Response 3: Thank you for bringing this to our attention. We acknowledge that this was an oversight on our part. The figures were created using Biorender, and we have now included a relevant statement in each legend to address this.

Point 4: I do not see the summary subsections that appear in each of the sections as appropriate, please delete.

Response 4: We agree with your suggestion to remove the summary subsections that appeared in each of the sections. While we believed that restating some of our key messages would help the reader, we acknowledge that it could also lead to unnecessary repetition. Therefore, we have deleted these sections as requested.

Point 5: Otherwise, this manuscript complies with journal guidelines and is suitable for publication after the above changes.

Response 5: Thank you once again for your kind comments. We appreciated your positive feedback, and we would like to confirm that we have addressed all of the points you raised and believe that the revisions have strengthened the manuscript.

Reviewer 4 Report

REVIEWER COMMENTS: "Breaking the vicious cycle: non-pharmacological strategies for managing sarcopenia in older adults with cardiovascular and metabolic diseases".

I want to thank the authors for the opportunity to review this work. I feel it could make contributions to healthcare area. It is a complete review that has a multidimensional approach in the management of sarcopenia in older adults with cardiovascular and metabolic diseases. However, it needs minor revisions in order to be improved.

There is an excess of bibliographical references. Perhaps some of them are unnecessary, for example in line 59, you use 3 references to define a single concept of sarcopenia. Review and use only necessary references.

Line 43: delete "...such as diabetes", since it is repeated below which conditions are included under the term metabolic diseases (MDs).

Line 87: delete the word modifiable from “Modifiable risk factors for sarcopenia are depicted in Figure 1”. The figure also includes non-modifiable factors.

Line 422: Typographic error in “….he occurrence…”

Author Response

Dear Reviewer,

Many thanks for your kind comments and the useful suggestions. We have addressed all of your individual points and made an effort to reduce the number of references. We agree with you that the number of references is large, but this is partly due to the multidisciplinary nature of the review. Nevertheless, we believe that it will be a very rich source for any academics, researchers, or health professionals who are interested in the area and wish to inform their practice. We did manage to reduce the number of references by 21. For any further reduction, we would have to start changing the meaning of several statements, and we are concerned that this may influence the quality of the manuscript. We hope this is acceptable for you.

Kind regards

Dr Theocharis Ispoglou

Round 2

Reviewer 1 Report

Non 

Author Response

Thank you for your review of our paper. I appreciate the time and effort you have dedicated to evaluating our work.

Reviewer 2 Report

The authors have made a good revision, now, I recommend to accept. 

N//A

Author Response

Thank you for your positive evaluation of our revised manuscript. We sincerely appreciate your time and effort in reviewing our work. We are delighted to hear that our revisions have met your expectations. Thank you once again for your valuable feedback and recommendation.